# Monitoring the Interfacial Polymerization of Piperazine and Trimesoyl Chloride with Hydrophilic Interlayer or Macromolecular Additive by In Situ FT-IR Spectroscopy

**DOI:** 10.3390/membranes10010012

**Published:** 2020-01-07

**Authors:** Xi Yang

**Affiliations:** Department of Polymer Science & Engineering, Zhejiang University, Hangzhou 310027, China; 11529019@zju.edu.cn; Tel.: +86-187-5810-1644

**Keywords:** interfacial polymerization, in situ FT-IR spectroscopy, thin-film composite membrane, nanofiltration membrane

## Abstract

The interfacial polymerization (IP) of piperazine (PIP) and trimesoyl chloride (TMC) has been extensively utilized to synthesize nanofiltration (NF) membranes. However, it is still a huge challenge to monitor the IP reaction, because of the fast reaction rate and the formed ultra-thin film. Herein, two effective strategies were applied to reduce the IP reaction rate: (1) the introduction of hydrophilic interlayers between the porous substrate and the formed polyamide layer, and (2) the addition of macromolecular additives in the aqueous solution of PIP. As a result, in situ Fourier transform infrared (FT-IR) spectroscopy was firstly used to monitor the IP reaction of PIP/TMC with hydrophilic interlayers or macromolecular additives in the aqueous solution of PIP. Moreover, the formed polyamide layer growth on the substrate was studied in a real-time manner. The in situ FT-IR experimental results confirmed that the IP reaction rates were effectively suppressed and that the formed polyamide thickness was reduced from 138 ± 24 nm to 46 ± 2 nm according to TEM observation. Furthermore, an optimized NF membrane with excellent performance was consequently obtained, which included boosted water permeation of about 141–238 (L/m^2^·h·MPa) and superior salt rejection of Na_2_SO_4_ > 98.4%.

## 1. Introduction

The interfacial polymerization (IP) method has been widely employed to fabricate polyamide-based membranes with a thin-film composite (TFC) structure, which include top polyamide layer, middle ultrafiltration support, and bottom non-woven fabric. TFC membranes are extensively used in the waste-water treatment field of reverse osmosis (RO) [1,2], nanofiltration (NF) [3], forward osmosis (FO) [4,5], and gas separation (GS). As it is reported in the literature, the IP process usually takes place at the interface of two immiscible solvents, such as the water/hexane interface. Typical reactive monomers of diamine and acid chloride monomers are separately dissolved in the solvents [6]. The diamine monomer is able to diffuse into the organic phase and subsequently reacts inside the organic phase near the interface, and then immediately forms the polyamide thin film in situ at the interface within several seconds [7]. As a consequence, the IP reaction has been successfully utilized to synthesize an ultra-thin and dense polyamide layer on the top surface on a porous substrate, which serves both as the storage location of the diamine monomer, and as the support of the formed ultra-thin polyamide film [8,9].

The IP reaction usually has a rapid reaction rate of about 10^4^ mol/(L·s), and the thickness of the formed ultra-thin film is normally less than 100 nm [10]. Therefore, it is a huge challenge to monitor the IP reaction in a real-time manner, as well as investigating the IP reaction kinetics directly on the porous substrates [10]. To the best of our knowledge, few methods have been developed to study the IP reaction, where the polyamide film is formed at the free water/oil interface. It is possible for one to terminate the IP reaction at different times, including determining the film mass [11], measuring the film thickness [12], and/or analyzing the reactant concentration [13,14,15]. Furthermore, the polyamide film formation at the free water/oil interface has also been observed by measuring the suspended drop interfacial tension [16], light reflection [16], and diffusion reflectance spectra [17]. Matthews et al. used Rutherford backscattering spectrometry (RBS) to study the polyamide growth dynamics, and established a relationship between the diffusion reflectance and the polyamide layer thickness [17]. Recently, Nowbahar et al. applied the microfluidic interferometry to measure the IP reaction kinetics of the m-phenylenediamine (MPD)/trimesoyl chloride (TMC) reaction system [10]. Although these techniques can be used to roughly estimate the IP reaction, they are still limited to detecting the IP reaction at the free water/oil interface, instead of directly on the porous substrate. Therefore, it is of great significance to develop effective strategies for monitoring the IP reaction, which takes place on the substrate in a real-time manner.

Herein, we report in situ Fourier transform infrared (FT-IR) spectroscopy as an effective instrument to monitor the IP reaction and the polyamide film growth on the porous substrate in real-time. In situ FT-IR spectroscopy is a powerful and well-established tool to investigate the reaction taking place at the solid/liquid interface [18,19], and to provide the real-time information about the IP reaction, such as mechanism and kinetics [20,21]. For example, Han et al. used a self-made FT-IR sample cell to monitor the photo-polymerization of alicyclic methacrylate hydrogel for controlled drug release [22]. Zimudzi et al. analyzed the Nafion film thickness [23], and quantified the carboxylic acid concentration of the polyamide-based membrane [24]. Moreover, it is reasonable to associate the absorbance intensity of characteristic band in the FT-IR spectra with the polyamide film thickness by mathematical equations [25,26,27,28]. Additionally, Ren et al. used FT-IR microscopy for IP reaction of microporous polymer film formed in the aqueous/organic interface for polyesterification by choosing the adequate phenol monomer [29]. Yang et al. reduced the IP reaction rate of piperazine (PIP) and trimesoyl chloride (TMC) reaction system to a certain extent by introducing interlayers between the porous substrate and the polyamide layer [30,31,32]. Tan et al. added the macromolecular additive of polyvinyl alcohol (PVA) in the aqueous solution of PIP and obtained the Turing structure [33]. Furthermore, in situ FT-IR spectroscopy is used to measure the IP reaction in real-time. The relationship between the IP reaction rate and the polyamide film thickness was successfully established for laying a foundation to optimize the NF membrane performance. The fabricated NF membranes exhibited boosted water permeation of about 141–238 (L/m^2^·h·MPa) and superior salt rejection of Na_2_SO_4_ > 98.4%. These improved NF membrane performances were attributed to the reduced polyamide layer thickness and the remaining high cross-linking degree of the internal structure of the polyamide layer.

## 2. Materials and Methods

### 2.1. Materials

Polysulfone (PSf) ultrafiltration membranes with a molecular weight cut-off (MWCO) of 50 kDa were used as the porous substrates, and were purchased from Ande Membrane Separation Technology and Engineering Co., Ltd. (Beijing, China). Dopamine hydrochloride (DA), tris-(hydroxymethyl) aminomethane (Tris), and *N*,*N*-bis(2-hydroxyethyl)glycine (Bicine) were obtained from Sigma-Aldrich (Shanghai, China). Polyethyleneimine (PEI, M_w_ ≈600 Da), tannic acid (TA), polyethyleneglycol (PEG, M_w_ = 20,000), polyvinylpyrrolidone (PVP, K23–27, M_w_ ≈24,000), polyvinyl alcohol (PVA 1788, M_w_ 75,000–80,000, 87%–89% hydrolyzed), and piperazine (PIP) were purchased from Aladdin Chemical Co., Ltd. (Shanghai, China). Trimesoyl chloride (TMC) was procured from Qingdao Benzo Chemistry Co., Ltd. (Qingdao, China). 2-Methylimidazole (Hmim) was purchased from Tokyo Chemical Industry Co., Ltd. (Tokyo, Japan). Other chemicals, including Zn (NO_3_)_2_·6H_2_O, *n*-hexane, ethanol, and inorganic salts (Na_2_SO_4_, MgSO_4_, MgCl_2_, CaCl_2_, and NaCl) were bought from Sinopharm Chemical Reagent Co., Ltd. (Shanghai, China). Ultra-pure water consumed in the experiments was directly generated from a lab water purification system (HYP-QX, Hangzhou, China).

### 2.2. Fabrication of Interlayers on the Porous Substrate

The porous substrates were cut into circular shape with a diameter of 47 mm and rinsed in deionized (DI) water and ethanol overnight. Various interlayers were fabricated on the porous substrate, including polydopamine (PDA)/polyethyleneimine (PEI), tannic acid (TA)/polyethyleneimine (PEI), and zeolitic imidazolate framework-8 (ZIF-8)/polyethyleneimine (PEI).

The PDA/PEI interlayer was constructed on the porous substrate, according to the reported literature. DA and PEI were dissolved in Tris-buffer (50 mM, pH = 8.5) at a mass ratio of 1:1, with a concentration of 2.0 mg/mL [30]. The porous substrate was immersed in the freshly prepared PDA/PEI solution for 1 h. After that, the PDA/PEI modified substrate was washed thoroughly by DI water and dried at room temperature for 1 h.

The TA/PEI interlayer was fabricated on the porous substrate via Michael addition or Schiff base reaction between the amino groups and quinone groups [31]. TA (2.0 mg/mL) was dissolved in a Bicine buffer solution (pH = 7.8), then PEI (2.0 mg/mL) was added in the solution. The substrate was immersed in the freshly prepared TA/PEI solution for 1 h, followed by washing procedure with DI water and drying at room temperature for 1 h of the TA/PEI modified substrate.

The ZIF-8/PEI interlayer was constructed on the porous substrate with the following steps [34,35]. The porous substrate was immersed in an equal volume of 0.1 mol/L Zn (NO_3_)_2_ aqueous solution and 0.4 wt % PEI aqueous solution for 0.5 h at 25 ± 2 °C. Next, the substrate was rinsed with DI water to remove the excess residual PEI molecules. Then, the porous substrate was immersed in 0.2 mol/L Hmim in *n*-hexane solution, which reacted with Zn^2+^ for 0.5 h. The as-prepared ZIF-8/PEI modified substrate was rinsed with DI water and dried at room temperature for further usage.

### 2.3. IP on the Substrates with Modified Interlayers

The porous substrates modified with as-formed interlayers were upper-side immersed in 10 mL aqueous solution of 2.0 g/L PIP for 10 min, ensuring the complete adsorption and infiltration of diamine solution on the interlayer surfaces and the substrate internal pores. A rubber roller was gently used to thoroughly dry the excess PIP solution on the top surface. Then, 10 mL of TMC solution with a concentration of 2.0 g/L in *n*-hexane was carefully poured on the substrates. Therefore, IP reaction was then consequently carried out for about 100 s of the optimum IP reaction time, to form the ultra-thin polyamide layers on top surfaces of the porous substrates. The resulted membranes were post-treated in an oven for 30 min at 60 °C, in order to further solidify and stabilize the formed polyamide layer.

### 2.4. IP on the Substrates with Macromolecular Additives

Macromolecular additives of PEG, PVP, and PVA were added into the aqueous solution of PIP. The aqueous solutions were prepared by mixing 1.5 g/L of macromolecular additive evenly with 2 g/L of PIP. The aqueous solutions were stirred at 25 ± 2 °C for 1 h and rested for 30 min to be uniformly mixed and to eliminate air bubbles. Then the IP reaction was then conducted on the porous substrates as per the method above. The resulting membranes were post-treated in an oven for 30 min at 60 °C, in order to further solidify and stabilize the formed polyamide layer.

### 2.5. Measuring IP Reaction by In Situ FT-IR Spectroscopy

Polyamide formation on the porous substrates was measured by in situ FT-IR spectroscopy (React 15, Mettler Toledo, Switzerland), with the scan range between 4000–650 cm^−1^, the maximum resolution of 4 cm^−1^, and the minimum detection time interval of 15 s. During the detection process, one sample of the porous substrate was fixed in a home-made reactor to ensure the upper-side made contact with the monomer solutions of PIP and TMC. Firstly, the FT-IR baseline was set up in the air environment, and then 10 mL of PIP solution (2.0 g/L) was poured onto the substrate surface for 10 min. The PIP solution was then poured off thoroughly and wiped by filter paper to remove the excess PIP solution on the surface. An optic probe was tightly contacted with the middle area of the substrate surface. Then, 10 mL of TMC solution (2.0 g/L) was poured on the substrate surface for conducting the IP reaction for a certain time. Data collection instantaneously began by the in situ FT-IR spectroscopy monitoring process. Subsequently, the FT-IR absorbance intensity was acquired and calculated from the polyamide film formation as a function of IP reaction time. In addition, measuring the polyamide layer thicknesses through FT-IR via Appendix A was only demonstrated for dense uniform films and, although this is not the case for IP-based membranes, it could still yield valid information on the IP process. In the calculation in the Appendix A, it is necessary to assume the polymer density to be 1.5, even though this is approximate, because the density is not constant throughout a PA film, nor throughout the interfacial polymerization reaction time.

### 2.6. Other Characterizations

Surface morphologies were observed by field emission scanning electron microscopy (FESEM, Hitachi, SU-8010, Japan) for the porous substrates and the formed polyamide layer. Topographies and surface roughness were measured by atomic force microscopy (AFM, MultiMode, Vecco, USA) in the tapping mode. Transmission electron microscopy (TEM, Hitachi 7650, Japan) was applied to observe the cross-sectional morphologies and to measure the thickness of polyamide layer by the microtomy sections. The samples were embedded in LR white resin (London Resin Company, Reading, UK), cut by ultramicrotome (Leica Microsystems, Wetzlar, Germany), and then mounted to copper grids. The crystalline structure of ZIF-8 was analyzed by an X-Ray diffractometer (XRD, 7000S/L, Shimadzu, Japan), using Cu Kα radiation, in the interval of 5° ≤ 2θ ≤ 50°, with simulated pattern was generated from CIF files of ZIF-8. Chemical structures were analyzed by attenuated total reflectance Fourier transform infrared spectrometry (ATR/FT-IR, Nicolet 6700, USA) and X-ray photoelectron spectrometry (XPS, Thermo Scientific, USA). Surface hydrophilicity was measured by a contact angle measurement system (Surface-Meter, OCA 200, China), and each sample was measured three times to get the average value and deviations. The surface zeta potential was evaluated by an electro kinetic analyzer (SurPASS Anton Paar, GmbH, Austria).

### 2.7. NF Membrane Performance Evaluation

NF membrane performance of the as-formed membranes were evaluated by a laboratory scale cross-flow flat module. The effective filtration area was 7.07 cm^2^ and the cross-flow rate was 30 L/h under the operation pressure of 0.6 MPa at room temperature for 1 h reaching the steady state. Various inorganic salts, including Na_2_SO_4_, MgSO_4_, MgCl_2_, CaCl_2_, and NaCl, were dissolved in DI water at a concentration of 1000 mg/L to prepare the feed solutions. The water permeation (*J_w_*, L/m^2^·h) and the salt rejection (*R*, %) were calculated by the following equations:(1)$$J_{w}=\frac{V}{A\cdot t}$$
where *V* (L), *A* (m^2^), and *t* (h) represent the filtered water volume, the effective membrane area, and the permeation time, respectively.
(2)$$R=\left(1-\frac{C_{p}}{C_{f}}\right)\times 100\%$$
where *C_p_* and *C_f_* (mg/L) are the feed and permeated solution concentrations, respectively, determined by an electrical conductivity meter (Mettler Toledo, FE30, Hangzhou, China) for average value as a result of three repeated measurements.

## 3. Results and Discussion

A typical IP reaction is usually finished in 0.5–2.0 min, because of the fast and uncontrollable IP reaction. In this work, hydrophilic interlayers and macromolecular additives were used to reduce the IP reaction rate for in situ FT-IR spectroscopy analyzing (as schematically shown in Figure 1). In addition, ATR/FT-IR spectra validated the formation of interlayers including PDA/PEI, TA/PEI, and ZIF-8/PEI on the porous substrates (Appendix A). As a result, the formed interlayers reasonably changed the surface morphology, surface hydrophilicity, and surface charge of the porous substrates (Appendix A).

Figure 2 and Figure 3 show the in situ FT-IR spectra monitoring the IP reaction on the porous substrates of C=O stretching vibration, specifically shown at 1640 cm^−1^. The absorbance intensities are also shown in Appendix A, as an indicator of the polyamide formation process. Here, I took the carbonyl vibration at 1640 cm^−1^. However, this was the carbonyl vibration of the amide, and as such there will be an error depending on the variation in chemical structure which was based on the following XPS analysis, because the polyamide structure was heterogeneous in the polyamide layer depth [36,37]. Figure 2 shows that the in situ FT-IR spectra monitored the IP reaction on the pristine PSf porous substrate, which nicely complied with the traditional growth principle. Usually, in situ FT-IR can be used to monitor a reaction for several hours; in this case of a fast IP reaction, the monitoring time was fixed between 0–300 s, with the minimum interval of 15 s. In other circumstances, the IP reaction rates are obviously repressed, in the condition of the introduction with hydrophilic interlayers and/or macromolecular additives. The reason for the repression phenomena could be possibly attributed to the slowed-down PIP diffusion rates, which were studied and acquired from the ultraviolet spectroscopy (UV-VIS) analyses for the 10 min diffusion time of PIP from water to hexane, as shown in Appendix A. As a result, the measured PIP diffusion rates are commendably in accordance with the decreased diffusion co-efficient reported in the literature [33], which reduced from about 10^−5^ to 10^−6^ cm^2^/s, as shown in Appendix A.

Furthermore, the reduced IP reaction rates also arose from the combination results of enhanced storage ability of PIP in the interlayers on the modified PSf substrate, and the enhanced interactions between the PIP and interlayers (Appendix A). In detailed analysis, the adhesive PDA/PEI and TA/PEI interlayers impeded the IP reaction rate and maintained a similar growth tendency compared with the IP reaction on the pristine PSf substrate. Additionally, polyamide growth on the ZIF-8/PEI interlayer obeyed the approximately painful and linear propensity, as the growth rate was nearly maintained in an unchanged manner during the IP reaction. The reason behind this was that ZIF-8/PEI is a high-porosity and inorganic interlayer, which restores and reduces PIP diffusion greatly. As for the doped polyamide layer with the macromolecular additives, polyamide growth was considerably repressed because of the low flow viscosity of macromolecular additives and their strong hydrogen-bonding interactions with the PIP monomer (as shown in Appendix A).

It is still a huge challenge to measure polyamide layer thickness, which formed in situ on the porous substrate. Previously, various methods were used to deal with this issue, including reducing the IP reaction rate [11] or stopping the IP reaction at various times and then determining the formed film mass or thickness [12]. V. Freger’s famous mathematical model predicts that the polyamide film thickness increases with the IP reaction time as t^1/2^, which is reported in the literature [36,37]. In this study, the in situ FT-IR absorbance intensities were related with the polyamide layer thickness. Figure 3 shows the growth rate of polyamide unitless thickness as a function of the IP reaction time. As a result, this method of FT-IR spectroscopy did not take variations in surface morphology of the polyamide into account, with this being another potential source of error on the thickness determination using FT-IR spectroscopy. Because there are questions regarding the vibration, the chemistry, the density, and the fact that in this study there was no validation on the FT-IR-based thickness, it would be better to use a unitless thickness in Figure 3. It can be seen that the IP reaction that took place on the pristine PSf substrate held the fastest film thickness increasing rate. In addition, the layer thickness of polyamide with the IP reaction time can be fitted as X = 24.8 t^1/2^, where X (nm) is polyamide thickness and t (s) is the IP reaction time (Appendix A). Furthermore, the growth of polyamide layers thickness are mathematically described as X = 7.3 t^1/2^, 4.0 t^1/2^, and 0.6 t on the various substrates modified with PDA/PEI, TA/PEI, and ZIF-8/PEI interlayers, respectively (as shown in Figure 3a,b and Appendix A). At the same time, when the macromolecular additives were added in the aqueous solution of PIP (Figure 3c,d and Appendix A), the growth of polyamide thickness corresponded to X = 5.4 t^2/3^, 3.3 t^2/3^, and 0.5 t, respectively, which implied the repressed IP reaction rate by the introduction of macromolecular additives in the aqueous solution of PIP.

In the aim of confirming the accuracy and reliability of the in situ FT-IR spectroscopy method for the determination of the thickness of the polyamide layer, the TEM technique was employed, as shown in Figure 4 and Appendix A. Figure 4 shows that the thickness of polyamide layers decreased noticeably with the introduction of interlayers and/or macromolecular additives. The ascertained thickness of polyamide layers obtained from the TEM images were at the sequence of 138 ± 24 nm (PSf) > 95 ± 7 nm (PEG) > 84 ± 17 nm (PVP) > 76 ± 19 nm (PVA) > 71 ± 3 nm (PDA/PEI) > 60 ± 6 nm (ZIF-8/PEI) > 46 ± 2 nm (TA/PEI). Furthermore, the as-formed polyamide layer surface morphologies and the roughness were observed and evaluated by the FESEM images (Appendix A) and the AFM images (Appendix A). The obtained results showed that the polyamide surface morphologies maintained the “nodular” structures on the pristine PSf substrate and changed on the substrates modified with various interlayers, because the IP reaction happened directly on the interlayers and had different surface morphology features. Furthermore, with the ever-increasing ability of PVA > PVP > PEG to retard the PIP diffusion rate, the doped polyamide layers with macromolecular additives could generated the Turing structure, as reported in the literature [33].

Moreover, XPS analyses were used to investigate the elemental contents near the polyamide layer surface (shown in Appendix A). With regard to the traditional formed polyamide membrane on the PSf substrate, the polyamide layer retained the conventional “sandwich-structure”, which meant the densest part was in the middle of the polyamide layer structure [36,37] and was loose on the top and bottom parts. In contrast, compared with the traditional polyamide structure, the polyamide layers formed on the pristine PSf substrates modified with various hydrophilic interlayers and led to the higher O/N ratio and lower cross-linking degree (shown in Appendix A), further verifying the looser top structures of XPS penetration depth less than 10 nm. Nevertheless, for doped polyamide layers with the macromolecular additives added in the aqueous PIP solution, the polyamide layer top structures were comparatively denser, with the lower O/N ratio and higher cross-linking degree (shown in Appendix A) [38].

NF performances of the as-prepared polyamide membranes were evaluated by laboratory cross-flow equipment. The membranes’ surface hydrophilicity were significantly improved by the introduction of interlayers and/or addition of macromolecular additives in the aqueous PIP solution, because of the potential changes of the polyamide surface groups. Moreover, polyamide membranes’ surface charge were evaluated by the zeta potential analyzer (shown in Appendix A). The results of their surface charges indicated that NF membranes with the hydrophilic modified interlayers had higher surface charges compared with the doped polyamide membranes with macromolecular additives in the PIP solution.

It was noticeable that the NF performances, including the water permeation and inorganic salts rejection rate, were usually the synergistic results of the polyamide layer thickness and their internal cross-linking degree. With the aim of obtaining the high membrane water permeation, the polyamide layer thickness was expected to be as thin as possible. Otherwise, for achieving the high inorganic salts rejection rate, the membrane internal high cross-linking degree was necessarily required. In our experiments, after the introduction of hydrophilic interlayers or adding of the macromolecular additives in the aqueous PIP solution, water permeation enhancements were noteworthy, which was in the order of ZIF-8/PEI > TA/PEI > PDA/PEI, with the hydrophilic interlayers on the modified porous substrates, and PVA > PVP > PEG for the doped polyamide membranes with macromolecular additives (as shown in Figure 5a).

Furthermore, inorganic salts’ rejection of these NF membranes were presented at the order of Na_2_SO_4_ > MgSO_4_ > MgCl_2_ > CaCl_2_ > NaCl (Appendix A), which is in accordance with the typical NF membranes carried with the surface negative charges. In particular, the high rejections of MgCl_2_ and CaCl_2_ for these doped polyamide membranes with macromolecular additives could be attributed to the “compactly packing” of the macromolecular additives in the polyamide layer structures. By taking into account the combination of size sieving and Donnan exclusion effects, the hydrated radius of ions was at the order of Mg^2+^ (0.428 nm) > Ca^2+^ (0.412 nm) > Na^+^ (0.358 nm), and thus the resulting rejection of NF membranes was in the sequence of MgCl_2_ > CaCl_2_ > NaCl [40,41]. Furthermore, the electrostatic interaction between SO_4_^2−^ and the polyamide negative surface was stronger than Cl^−^, leading to the high rejections of Na_2_SO_4_ and MgSO_4_ (exceeding above 97.2%). In summary, the as-prepared NF membranes provided a great opportunity to break out the “trade-off” behavior of the traditional polyamide membranes, showing high water permeation, and maintaining a superior inorganic salt rejection rate, thus having practical use in industry (as shown in Figure 5b).

## 4. Conclusions

To sum up, in situ FT-IR spectroscopy was for the first time employed to measure the IP reaction on the porous substrate. The results successfully validated the “depressed-effect” of both the introduced interlayers on the pristine PSf substrate and macromolecular additives in the PIP solution during the IP reaction. Though the exact forming mechanism of the dissimilar growing types of IP reaction were still not clear, the in situ FT-IR spectroscopy paved a new way in which to thoroughly comprehend the IP reaction mechanism. Furthermore, it could be utilized to tailor the superior performing thin-film composite membrane, consequently having significant and practical usage in the water treatment industry.

## Figures and Tables

**Figure 1 membranes-10-00012-f001:**
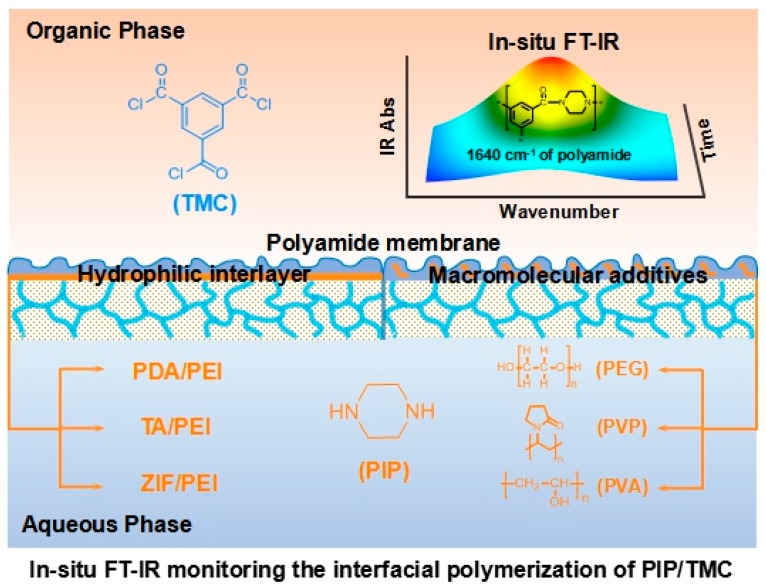
In-situ Fourier transform infrared (FT-IR) spectroscopy monitoring the interfacial polymerization (IP) reaction of piperazine (PIP)/trimesoyl chloride (TMC) by the introduction of hydrophilic interlayers on the pristine polysulfone (PSf) substrate or macromolecular additives in the aqueous solution of PIP. PDA: polydopamine (PDA), PEI: polyethyleneimine, TA: tannic acid, ZIF: zeolitic imidazolate framework, PEG: polyethyleneglycol, PVP: polyvinylpyrrolidone, PVA: polyvinyl alcohol.

**Figure 2 membranes-10-00012-f002:**
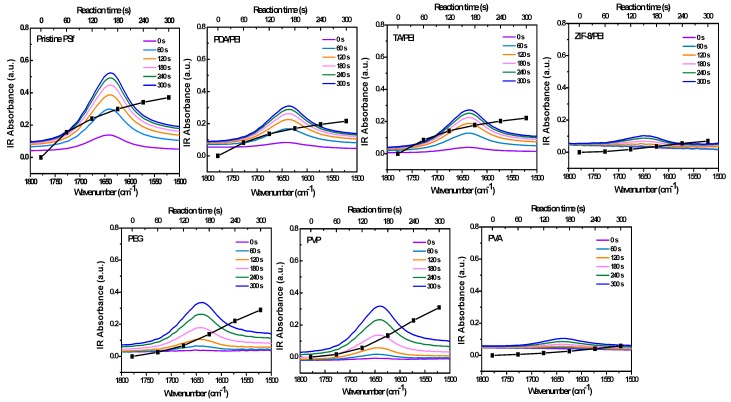
The in situ FT-IR spectra and absorbance intensities growth of polyamide formation process on porous substrate, which were characterized by the C=O stretching vibration at 1640 cm^−1^.

**Figure 3 membranes-10-00012-f003:**
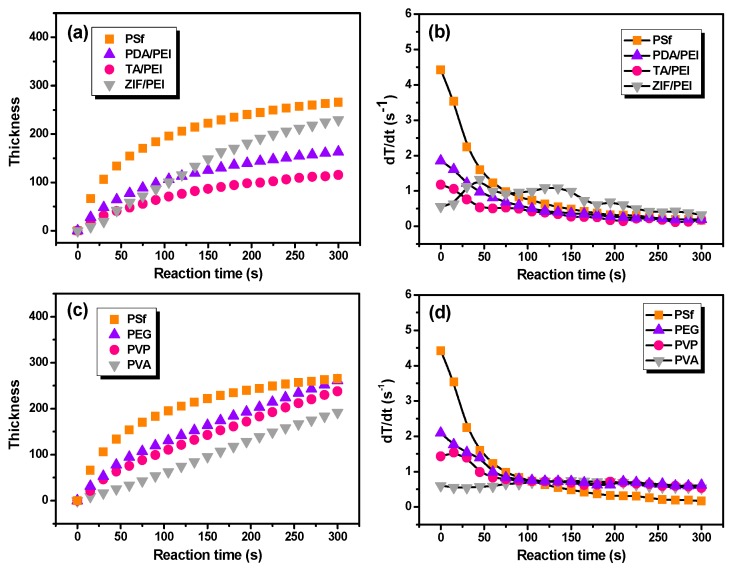
The growth of calculated polyamide layer thickness and the correspondingly polyamide layer thickness growth rate, as a function of IP reaction time including (**a**,**b**) IP reaction taking place on the porous substrates with various modified interlayers, (**c**,**d**) the doped polyamide layer with the macromolecular additives in the aqueous solution of PIP.

**Figure 4 membranes-10-00012-f004:**
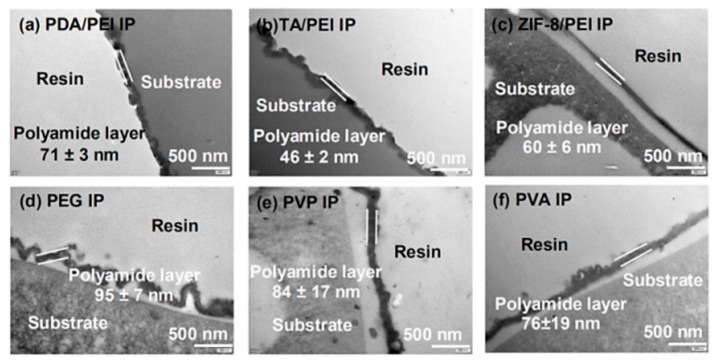
Transmission electron microscopy (TEM) observation of thickness of the formed polyamide layer on the (**a**–**c**) PSf substrates with various modified interlayers and (**d**–**f**) the doped polyamide layer with the macromolecular additives in the aqueous solution of PIP.

**Figure 5 membranes-10-00012-f005:**
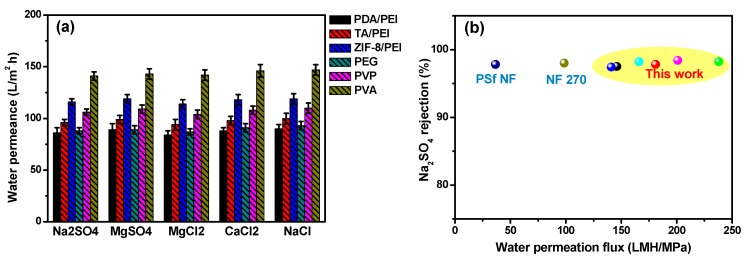
(**a**) Water permeation of the as-prepared NF membranes with the hydrophilic modified interlayers on the PSf substrates or the addition of macromolecular additives in the aqueous PIP solution (test conditions: inorganic salts concentration = 1000 mg/L, pH = 6.5, 25 °C, applied pressure = 0.6 MPa and cross-flow rate = 30 L/h); (**b**) the comparison of the NF performance with other NF membranes in the reported literature [30,39].

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
