# Peer review of "Monitoring the Interfacial Polymerization of Piperazine and Trimesoyl Chloride with Hydrophilic Interlayer or Macromolecular Additive by In Situ FT-IR Spectroscopy"

_membranes, 2020, doi:10.3390/membranes10010012_

Round 1
Reviewer 1 Report
In the manuscript entitled “Monitoring the Interfacial Polymerization of Piperazine and Trimesoyl Chloride with Hydrophilic Interlayer or Macromolecular Additive by in-situ FT-IR Spectroscopy” by Xi Yang, the author used FT-IR to monitor the formation of piperazine-based polyamide layers onto various surface-modified polysulfone supports to produce nanofiltration (NF) membranes. These resulting membranes were thoroughly characterized with standard methods and their filtration performance was evaluated. Overall, this study presents a significant amount of experimental work. Nevertheless, I found some inconsistencies between the main conclusions of the study and how the supporting evidence is presented (see detailed comments below). Therefore, the manuscript requires significant revisions to address these issues before it could be accepted for publication.
According to the author the two main results are:
1. FT-IR is established as a technique to “monitor the IP reaction and the polyamide film growth on the porous substrate in real-time”.
This was partially accomplished. The author demonstrated that FT-IR is sensitive to the appearance of polyamide. However, from the discussion it is not evident whether FT-IR is conclusively a reliable method for determining film thickness quantitatively for different membranes. The IP reaction was monitored with FT-IR until 300 s, but from Figure 2 it appears that this was not a long enough reaction time for all membranes (e.g., PEG or PVP). The author should discuss the sensitivity of the fitting curve to how long the reaction was monitored.
Figure 3 is also very confusing because the predicted film thickness values at 300 s are clearly different than the values observed from the TEM cross-sections. Only then I realized that the membranes of the first row were made with a reaction time of 100 s (Section 2.4). For clarification, this should be explicitly shown in Figure 3 and emphasized in the discussion. Similarly, the calculated thickness values from FT-IR data should be listed and explicitly compared with the TEM values. I believe this is one of the strong points of the manuscript, but it does not come across clearly.
One significant shortcoming is that there is no explanation as to why the author chose 100 s as the reaction time to prepare the membranes and whether this FT-IR method would predict film thickness at any other reaction times? Examples of one or two of the membranes at different reaction times would provide strong evidence for this method.
There is no specification regarding the reaction times for the membranes in the second row. Was the reaction time also 100 s? If not, why?
Why was the FT-IR scan range so broad from 4000-650 cm-1 if only the peak at 1640 cm-1was relevant? Would a shorter scan range (i.e., scanning interval) result in more accurate measurements?
2. Membranes with better performance were produced because the polyamide thickness was reduced by using interlayers or additives that slowed down the IP.
Figure 5c is grossly misleading by comparing rejections of different salts (“apples and oranges”) and needs to be modified. It implies that the membranes evaluated here completely outperform RO membranes, which is incorrect. The membranes in this study should only be compared to equivalent piperazine-based NF membranes. Salt rejection of RO membranes is >99% NaCl while the membranes here had a NaCl rejection of ~50%.
Furthermore, the author claims that performance was improved by reducing thickness, yet the performance values of the base membrane on the pristine polysulfone support was not reported. And even when comparing the other six membranes a correlation between film thickness and performance is not established.
For an even more complete analysis, the author should compare these membranes to commercial ones like NF270. As an interesting point, the polyamide layer of NF270 is already quite thin (~40 nm), which is very different than the thickness obtained for the base membrane in this study.
This section of the manuscript needs to be revised thoroughly.
Other comments and questions:
3. Line 184 and throughout the manuscript: The term “nascent” is used incorrectly. I believe the author means “pristine” or “unmodified” to describe the polysulfone support without interlayers.
4. Lines 30–36: These three sentences need revision because they are very confusing.
5. Lines 40–41: What is the reference for this reaction rate?
6. Figure 1: The placement of the labels “Hydrophilic interlayer” and “Macromolecules” on the location of the polysulfone support is confusing.
7. Line 187 & Figure S6: It is not clear when exactly the samples were taken to determine the diamine diffusion rates?
8. The way in which the thickness is drawn in Figure S10 appears very inaccurate.
9. Line 208: What is the meaning of “approximately painful and linear propensity”?
10. Lines 237–238: This statement is incorrect. The features observed on the polyamide surface are not of the same size as the polysulfone nodules. They would have formed even in the absence of the support (assuming the same reaction conditions). That is the reason why there are also similar features in the membranes that contain interlayers, where the polysulfone nodules are irrelevant.
Author Response
Response to Reviewer 1 Comments
Point 1: FT-IR is established as a technique to “monitor the IP reaction and the polyamide film growth on the porous substrate in real-time”.
This was partially accomplished. The author demonstrated that FT-IR is sensitive to the appearance of polyamide. However, from the discussion it is not evident whether FT-IR is conclusively a reliable method for determining film thickness quantitatively for different membranes. The IP reaction was monitored with FT-IR until 300 s, but from Figure 2 it appears that this was not a long enough reaction time for all membranes (e.g., PEG or PVP). The author should discuss the sensitivity of the fitting curve to how long the reaction was monitored.
Figure 3 is also very confusing because the predicted film thickness values at 300 s are clearly different than the values observed from the TEM cross-sections. Only then I realized that the membranes of the first row were made with a reaction time of 100 s (Section 2.4). For clarification, this should be explicitly shown in Figure 3 and emphasized in the discussion. Similarly, the calculated thickness values from FT-IR data should be listed and explicitly compared with the TEM values. I believe this is one of the strong points of the manuscript, but it does not come across clearly.
One significant shortcoming is that there is no explanation as to why the author chose 100 s as the reaction time to prepare the membranes and whether this FT-IR method would predict film thickness at any other reaction times? Examples of one or two of the membranes at different reaction times would provide strong evidence for this method.
There is no specification regarding the reaction times for the membranes in the second row. Was the reaction time also 100 s? If not, why?
Why was the FT-IR scan range so broad from 4000-650 cm-1 if only the peak at 1640 cm-1 was relevant? Would a shorter scan range (i.e., scanning interval) result in more accurate measurements?
Response 1: As the reviewer kindly suggested, the in-situ FT-IR spectroscopy was carried out for 10 min, with the minimum interval of 15 s, because of the limitation of the instrument. It is true that the 300 s detection time maybe not enough for all IP reaction in my experiments.
However, for comparing with the fast IP reaction happened on the pristine PSf substrate (usually less than 120 s), the exhibited data was between the range of 0-300 s, and 100 s is the optimum IP reaction time according to our previous study published on Langmuir.
In addition, for the techniques limitation, the polyamide layer thickness calculation and the observations from TEM images maybe not be accurate enough, which is a proposed idea for investigating the polyamide layer thickness by different methods.
Furthermore, in order to enlarge the peak change in the FT-IR spectra, the peak at 1640 cm-1 exhibiting the polyamide formation are shown in Figure 2.
Point 2: Membranes with better performance were produced because the polyamide thickness was reduced by using interlayers or additives that slowed down the IP.
Figure 5c is grossly misleading by comparing rejections of different salts (“apples and oranges”) and needs to be modified. It implies that the membranes evaluated here completely outperform RO membranes, which is incorrect. The membranes in this study should only be compared to equivalent piperazine-based NF membranes. Salt rejection of RO membranes is > 99% NaCl while the membranes here had a NaCl rejection of ~50%.
Furthermore, the author claims that performance was improved by reducing thickness, yet the performance values of the base membrane on the pristine polysulfone support was not reported. And even when comparing the other six membranes a correlation between film thickness and performance is not established.
For an even more complete analysis, the author should compare these membranes to commercial ones like NF270. As an interesting point, the polyamide layer of NF270 is already quite thin (~ 40 nm), which is very different than the thickness obtained for the base membrane in this study.
Response 2: Thanks for the reviewer’s valuable advice. Figure 5 has been modified in the revised manuscript and removing the data of RO membranes. In addition, water permeance and salt rejection of NF 270 and PSf NF are also added.
Line 184 and throughout the manuscript: The term “nascent” is used incorrectly. I believe the author means “pristine” or “unmodified” to describe the polysulfone support without interlayers.
The term “nascent” has been replaced with “pristine”.
Lines 30–36: These three sentences need revision because they are very confusing.
These three sentences have been modified.
Lines 40–41: What is the reference for this reaction rate?
The reference [10] has been added.
Figure 1: The placement of the labels “Hydrophilic interlayer” and “Macromolecules” on the location of the polysulfone support is confusing.
The location of labels have been modified in the Figure 1.
Line 187 & Figure S6: It is not clear when exactly the samples were taken to determine the diamine diffusion rates?
The diamine diffusion rate from water to hexane determine time is 10 min, the same time as the immersion time for substrates.
The way in which the thickness is drawn in Figure S10 appears very inaccurate.
The polyamide layer thickness shown in Figure S10 has been adjusted.
Line 208: What is the meaning of “approximately painful and linear propensity”?
For the growth rate is nearly maintained unchanged, during the IP reaction.
Lines 237–238: This statement is incorrect. The features observed on the polyamide surface are not of the same size as the polysulfone nodules. They would have formed even in the absence of the support (assuming the same reaction conditions). That is the reason why there are also similar features in the membranes that contain interlayers, where the polysulfone nodules are irrelevant.
The obtained results shown that the polyamide surface morphologies maintain the “nodular” structures on the pristine PSf substrate and changed on the substrates modified with various interlayers, because of the IP reaction happened directly on the interlayers and have the different surface morphology features.

Reviewer 2 Report
The paper investigates the influences of different membrane modification coatings on the interfacial polymerization process. The quality of the writing is ok and the visual quality of the figures is excellent. However, I have serious questions with the setup of the experiments and the data interpretation. I question the scientific correctness of the FTIR experiments (see below). In addition, there is no real integration of the data of the different experiments to form the overall bigger story. As there are quite some serious issues, I don’t think the quality of this paper is sufficient and it should therefore be rejected. I outlined my major areas of concern in more detail below and at the end there is a part with smaller, more specific remarks.
Major concerns
In my opinion the materials and methods section is not consistent nor detailed enough. (1) in certain occasions a rubber roller is used to remove the excess organic solution whereas in other cases filter paper is used. IP is a very sensitive process, both to the reaction conditions but also to how it is performed in practice, mixing two types of practicalities basically makes comparing results impossible. (2) a second example of this can be found it the fact that in 2.3 there is no curing step apparently whereas there is one in 2.4. Again, results can thus not be compared. Even more worrying is the fact that it is also not clearly described how the reference membrane was prepared, with or without annealing?? (3) In 2.4 it is mentioned how the aqueous solution was prepared. In 2.3 it is not mentioned. Were they prepared the same or not? (4) the contact angle measurements are not described in the materials and methods. One is thus left wondering if e.g. the standard deviations on these measurements are from contact angle measurements on different spots of the membrane or if this is just the machine error on the measurement on one spot. Of course the former is the one that is much more valuable as this is the only way to estimate the variability in the application of the technique to these specific samples. (5) Also the FTIR measurements are in my opinion not described in enough detail. Was a baseline taken of the composition of the membrane before addition of the organic phase or was a full spectrum measured at any time?
I have major issues with the use of FTIR spectroscopy for thickness determination. The authors use this technique and its associated equations while referring to one publication from 1985 which demonstrates that this can be used to estimate the thickness of layers of polystyrene with varying thickness on a fixed thickness layer of polyurethane. This 1985 study thus uses 2 preformed polymers and varies the thickness of one dense layer on top of a fixed thickness dense layer of the other polymer. I really don’t think you can translate it to polyamide based on IP for the following reasons:
(1) The refractive index of PA is taken at 1.5 without reference. Where does this number come from? The formation of PA is not ‘a linear process’ of increasing thickness. It is commonly accepted that it forms a looser layer first which then densifies. As such it cannot be that the refractive index is a constant in time. Moreover I also don’t believe that the refractive index is constant when comparing samples with different crosslinking degree (and you demonstrate that the crosslinking degree is varying using the XPS). That means it is not physically correct to use this method / these equations. In the original paper from 1985 that you refer to the layer material and density is fully constant and thus so is the refractive index.
(2) In the report from 1985 the chemistry of the materials used is constant. As such it is thus very well possible to correlate the amount of material to the intensity of a specific vibration of the material. However, for PA prepared by IP the chemical structure is not necessarily constant during the time of the IP and it definitely is not constant when comparing samples prepared by different conditions (and you prove that they are not the same from your XPS!!!). As such it is again not possible to calculate thicknesses from FTIR vibrations. Let me illustrate this with a fictional example. If in one case you would have a fully crosslinked PA and you compare it to a sample where one of every 3 acid chloride groups are simply hydrolyzed to –COOH. Let’s now assume they both form layers of equal thickness vs time. Yet in your experiment it would look like in the hydrolysis case the layer is forming slower and remains thinner because much 1 out of 3 acid chloride groups goes to –COOH instead of an amide.
What I also find worrying in the FTIR experiment is that you see from figure S5 that there are also other peaks forming and that there are differences between the different samples with respect to which peaks arise and with which magnitude. This contains valuable information which is not analyzed/discussed at all.
The common way to describe the amount of permeate one obtains in nanofiltration/reverse osmosis research is permeance, not flux. Please change all to permeance, these numbers allow easy comparison between papers. Moreover you even mix flux and permeance throughout the manuscript.
On line 184 you mention that the nascent PSf case clearly shows the 3-phase process. Personally I don’t see how this is reflected in that curve? I think you should find a way to show this visually. Is it even possible to see this when you use a technique with only one sampling per minute?
One of the interlayers you prepare is composed of ZIF-8/PEI. If you prepare anything that relates to inorganic porous crystalline materials (be it either membranes or catalysts etc.) you should prove that the material that you intend to form is actually formed. In this case, you can really not talk about ZIF-8 without also showing an x-ray diffraction pattern that is matched with the theoretical pattern for ZIF-8. Alternative experiments that could also be beneficial are nitrogen sorption or STEM-EDX. (In figure 4, I don’t see the ZIF-8 layer?)
With respect to figure 5, I find this figure very confusing and difficult to read. (1) there is no significant difference in flux when comparing the different salts for one given membrane. I would therefore eliminate most of the salts and only show one or max two. Please also show the rejection for each of the salts that you put in the manuscript as this is crucal information. (2) I don’t understand why you don’t put the left and middle figure together? That would make the data much more easy to compare. Moreover don’t forget to include the nascent PSf data. (3) the left and middle figure are both showing flux whereas the right one shows permeance, please put them all in permeance. (4) You took these data from reference 39. This is a paper on TFN membranes with GO embedded in the PA layer. As such a lot of the data points are concerning TFN membranes with GO/TiO2/SiO2. These are not relevant to your work. Please prepare your own plot with relevant data (i.e. data on TFC membranes prepared via all kinds of different methods/conditions). Moreover, did you just leave of the data point of the ‘this work’ of reference 39? Finally these data definitely also require labels.
I don’t find sufficient effort in the discussion to analyze potential correlations between the different surface chemistries <> potential hypothesis on how this affects membrane formation <> the membrane performance and <> the membrane characterization. This is crucial to make a manuscript like this valuable.
Smaller more specific remarks
Abstract line 22, permeance is not L.m².h/MPa but L/(m².h.MPa).
In the introduction, please structure the part on the IP process. You mention several things twice and the explanation is not structured going back and forth in the IP process. Moreover it contains errors. E.g. the diamine does not react near the organic phase side, it is commonly accepted that it reacts inside the organic phase near the interface. In addition, the self-limiting character of the IP that is the cause for the formation of thin layers is not explained sufficiently nor clearly.
Line 27; ‘…the interfacial polymerization method…’
Line 28; remove typical
Line 30; PA-TFC membranes are also being explored for GS, please add this.
Line 32; add ‘typical’, because also other monomer types are used in IP, e.g. alcohols etc.
Line 36; as a consequence not consequent
Line 57; ‘…we report in-situ…’ (remove the ‘the’)
Line 137; ‘It was then poured off the PIP solution’ this is not correct English.
Line 149; please specify the thickness of the microtomy sections.
Why is the Nascent PSf figure added twice to figure 2? This is not necessary.
Section 2.7 does not describe how it was determined that the membranes were measured at steady state. This is crucial for comparing membranes.
Author Response
Response to Reviewer 2 Comments
Point 1: In my opinion the materials and methods section is not consistent nor detailed enough. (1) in certain occasions a rubber roller is used to remove the excess organic solution whereas in other cases filter paper is used. IP is a very sensitive process, both to the reaction conditions but also to how it is performed in practice, mixing two types of practicalities basically makes comparing results impossible. (2) a second example of this can be found it the fact that in 2.3 there is no curing step apparently whereas there is one in 2.4. Again, results can thus not be compared. Even more worrying is the fact that it is also not clearly described how the reference membrane was prepared, with or without annealing?? (3) In 2.4 it is mentioned how the aqueous solution was prepared. In 2.3 it is not mentioned. Were they prepared the same or not? (4) the contact angle measurements are not described in the materials and methods. One is thus left wondering if e.g. the standard deviations on these measurements are from contact angle measurements on different spots of the membrane or if this is just the machine error on the measurement on one spot. Of course the former is the one that is much more valuable as this is the only way to estimate the variability in the application of the technique to these specific samples. (5) Also the FTIR measurements are in my opinion not described in enough detail. Was a baseline taken of the composition of the membrane before addition of the organic phase or was a full spectrum measured at any time?
Response 1: (1) For substrates modified with interlayers, the rubber roller is useful to remove the surface diamine solution, and for other substrates, drying in air for 10 min can reach the similar effect. (2) The preparation procedures of annealing is the same and revised as the reviewer suggested. (3) Section 2.3 and 2.4, the NF membranes were prepare as the same procedure. (4) The contact angle measurements are re-described and each samples were measured for three times and get the average value and deviations. (5) The FT-IR baseline was set up in the air environment and 10 mL of PIP solution (2.0 g/L) was poured onto the substrate surface for 10 min, thus it is before addition of the organic phase.
Point 2: I have major issues with the use of FTIR spectroscopy for thickness determination. The authors use this technique and its associated equations while referring to one publication from 1985 which demonstrates that this can be used to estimate the thickness of layers of polystyrene with varying thickness on a fixed thickness layer of polyurethane. This 1985 study thus uses 2 preformed polymers and varies the thickness of one dense layer on top of a fixed thickness dense layer of the other polymer. I really don’t think you can translate it to polyamide based on IP for the following reasons:
(1) The refractive index of PA is taken at 1.5 without reference. Where does this number come from? The formation of PA is not “a linear process” of increasing thickness. It is commonly accepted that it forms a looser layer first which then densifies. As such it cannot be that the refractive index is a constant in time. Moreover I also don’t believe that the refractive index is constant when comparing samples with different cross-linking degree (and you demonstrate that the cross-linking degree is varying using the XPS). That means it is not physically correct to use this method / these equations. In the original paper from 1985 that you refer to the layer material and density is fully constant and thus so is the refractive index.
(2) In the report from 1985 the chemistry of the materials used is constant. As such it is thus very well possible to correlate the amount of material to the intensity of a specific vibration of the material. However, for PA prepared by IP the chemical structure is not necessarily constant during the time of the IP and it definitely is not constant when comparing samples prepared by different conditions (and you prove that they are not the same from your XPS!!!). As such it is again not possible to calculate thicknesses from FT-IR vibrations. Let me illustrate this with a fictional example. If in one case you would have a fully cross-linked PA and you compare it to a sample where one of every 3 acid chloride groups are simply hydrolyzed to -COOH. Let’s now assume they both form layers of equal thickness vs time. Yet in your experiment it would look like in the hydrolysis case the layer is forming slower and remains thinner because much 1 out of 3 acid chloride groups goes to -COOH instead of an amide. What I also find worrying in the FTIR experiment is that you see from figure S5 that there are also other peaks forming and that there are differences between the different samples with respect to which peaks arise and with which magnitude. This contains valuable information which is not analyzed/discussed at all.
Response 2: (1) Thanks to the reviewer’s kindly advice and carefully reading and analyzing the reference of the polyamide layer thickness calculation in the article. (2) Many thanks to the reviewer’s dedicated spirit for science, showing a very interesting and useful example, appropriate questions about the cross-linking degree and polyamide layer thickness, which I still need to understand for longer time. I want to propose that, for the techniques limitation, the polyamide layer thickness approximately calculation by reference and the observations from TEM images maybe not be accurate enough, which is a proposed idea for investigating the polyamide layer thickness by different methods. Furthermore, in order to enlarge the peak change in the FT-IR spectra, the peak at 1640 cm-1 exhibiting the polyamide formation are shown in Figure 2 and Figure S5.
Point 3: The common way to describe the amount of permeate one obtains in nanofiltration/reverse osmosis research is permeance, not flux. Please change all to permeance, these numbers allow easy comparison between papers. Moreover you even mix flux and permeance throughout the manuscript.
Response 3: The “flux” has been changed to “permeance” in the revised manuscript.
Point 4: On line 184 you mention that the nascent PSf case clearly shows the 3-phase process. Personally I don’t see how this is reflected in that curve? I think you should find a way to show this visually. Is it even possible to see this when you use a technique with only one sampling per minute?
Response 4: The 3-phase process is shown in the modified Figure 2.
Point 5: One of the interlayers you prepare is composed of ZIF-8/PEI. If you prepare anything that relates to inorganic porous crystalline materials (be it either membranes or catalysts etc.) you should prove that the material that you intend to form is actually formed. In this case, you can really not talk about ZIF-8 without also showing an x-ray diffraction pattern that is matched with the theoretical pattern for ZIF-8. Alternative experiments that could also be beneficial are nitrogen sorption or STEM-EDX. (In figure 4, I don’t see the ZIF-8 layer?)
Response 5: The XRD and EDX are shown in the revised supporting information.
Figure S3. XRD spectra of the synthesized ZIF-8 and ZIF-8/PEI.
Figure S4. FESEM and EDX images of the ZIF-8/PEI modified PSf substrate.
Point 6: With respect to figure 5, I find this figure very confusing and difficult to read. (1) there is no significant difference in flux when comparing the different salts for one given membrane. I would therefore eliminate most of the salts and only show one or max two. Please also show the rejection for each of the salts that you put in the manuscript as this is crucal information. (2) I don’t understand why you don’t put the left and middle figure together? That would make the data much more easy to compare. Moreover don’t forget to include the nascent PSf data. (3) the left and middle figure are both showing flux whereas the right one shows permeance, please put them all in permeance. (4) You took these data from reference 39. This is a paper on TFN membranes with GO embedded in the PA layer. As such a lot of the data points are concerning TFN membranes with GO/TiO2/SiO2. These are not relevant to your work. Please prepare your own plot with relevant data (i.e. data on TFC membranes prepared via all kinds of different methods/conditions). Moreover, did you just leave of the data point of the “this work” of reference 39? Finally these data definitely also require labels.
Response 6: (1) Rejections for other salts are shown in Table S7 in the supporting information. (2) Figure 5a and 5b has been combined, and pristine PSf NF data is added. (3) They have been all expressed with permeance. (4) Figure 5c has been modified, in the revised manuscript and removing the data of RO membranes. In addition, water permeance and salt rejection of NF 270 are also added.
Point 7: I don’t find sufficient effort in the discussion to analyze potential correlations between the different surface chemistries potential hypothesis on how this affects membrane formation the membrane performance and the membrane characterization. This is crucial to make a manuscript like this valuable.
Response 7: It can be known that the different surface morphology may lead to different membrane performance. Thus the high water permeance of the NF membrane which is mixed PEG, PVP, PVA is attributed to the change of the top layer surface morphology. To my point of view, hydrophilicity, cross-linking (compactness) as well as thickness will affect membrane water permeance.
Point 8: Smaller more specific remarks
Abstract line 22, permeance is not L.m².h/MPa but L/(m².h.MPa).
In the introduction, please structure the part on the IP process. You mention several things twice and the explanation is not structured going back and forth in the IP process. Moreover it contains errors. E.g. the diamine does not react near the organic phase side, it is commonly accepted that it reacts inside the organic phase near the interface. In addition, the self-limiting character of the IP that is the cause for the formation of thin layers is not explained sufficiently nor clearly.
Line 27; ‘…the interfacial polymerization method…’
Line 28; remove typical
Line 30; PA-TFC membranes are also being explored for GS, please add this.
Line 32; add ‘typical’, because also other monomer types are used in IP, e.g. alcohols etc.
Line 36; as a consequence not consequent
Line 57; ‘…we report in-situ…’ (remove the ‘the’)
Line 137; ‘It was then poured off the PIP solution’ this is not correct English.
Line 149; please specify the thickness of the microtomy sections.
Why is the Nascent PSf figure added twice to figure 2? This is not necessary.
Section 2.7 does not describe how it was determined that the membranes were measured at steady state. This is crucial for comparing membranes.
Response 8: Unit of “permeance” has been modified to L/(m2·h·MPa). In the introduction part, “TFC membranes are extensively used in the waste-water treatment field of reverse osmosis (RO), nanofiltration (NF), forward osmosis (FO) and gas separation (GS). As it reported in the literatures, IP process usually takes place at the interface of two immiscible solvents, such as water/hexane interface. Typical reactive monomers of diamine and acid chloride monomers are dissolved in the solvents, separately. The diamine monomer is able to diffuse into the organic phase and subsequently reacts inside the organic phase near the interface, and then in-situ immediately form the polyamide thin film at the interface within several seconds.”
The ‘as a consequence’ has been modified. Other small mistakes have been modified according to the reviewer’s kindly advice. One of the pristine PSf figure has been deleted. Section 2.7 re-described “the NF membrane performance of the as-formed membranes were evaluated by a laboratory scale cross-flow flat module, under the operation pressure of 0.6 MPa at room temperature for 1 h reaching the steady state.”

Round 2
Reviewer 1 Report
In my previous report, I mentioned two significant issues that the author needed to correct before this manuscript could be accepted for publication:
1. According to the author, the main objective of this manuscript is to “monitor the IP reaction and the polyamide film growth on the porous substrate in real-time”. I recognized that Figure 2 showed that FT-IR was qualitatively sensitive to both the appearance of a polyamide layer on the support and to the effects of slowing down the reaction with the use of interlayers and additives. However, I was not entirely convinced that this data could be used to measure the thickness of the layers, and even less predict or describe the relative performance of the resulting membranes.
The revised manuscript did not address any of these concerns, and in my opinion the same issues remain:
1a. The results did not provide clear evidence of whether FT-IR was capable of reliably measuring the thickness of the layers under different reaction conditions. I was also expecting to see a better comparison (e.g., in a table) of the thickness values obtained with FT-IR vs. those obtained from the TEM images, as well as a detailed discussion of the shortcomings of each technique.
Also, in order to validate the thickness calibration curves, images (or some other evidence) would be necessary at different reaction times to prove that the FT-IR values are truly tracking the growth of the layers. For example, if the standard reaction shown in the first panel of Figure 2 is allowed to proceed to 300 s, would the resulting polyamide layer be ~250 nm?
1b. There is no evidence or proper discussion of how this FT-IR method can actually help in the preparation of membranes with improved properties. The FT-IR data neither was used to optimize any of the membranes, nor provided essential information beyond the two qualitative aspects that I mentioned before. The change in the amine diffusivities was measured with UV-Vis and the relative thickness differences were shown with TEM images. In other words, the same membrane results would have been obtained without using FT-IR.
Of course, it would be a significant result if this FT-IR method could predict the characteristics of the resulting polyamide layer, even if it is just the film thickness, but for this it would be necessary to establish and verify proper calibration curves.
1c. The revised version of Figure 2 introduced (in the first graph panel) a division of three reaction stages based on the mathematical modeling of the IP reaction by Freger and Srebnik (Ref. 36). However, I do not see any evidence that the FT-IR curve corresponds to these theoretical stages. The current data and experiments are not enough to investigate the existence of these three stages. Even the experimental work by Freger (Ref. 37) or any other study, as far as I know, have not shown this conclusively. Furthermore, such a formation mechanism would still be true at least for some of the other cases with different reaction rates. It would be necessary for the author to explain what exactly is FT-IR measuring in the other cases with respect to the formation of the polyamide layer.
2. My second point was related to how the comparison of the membrane performance results were presented in a very misleading way in the original version of Figure 5. The updated version of Figure 5 made the necessary corrections and it addresses my concerns. The only minor comment is that the leftover red arrow in Figure 5b should be removed.
I would be inclined to support this manuscript if the author were to shift the main focus from a paper about FT-IR as a method to monitor the IP reaction to a paper about the fabrication of membranes with improved performance by modifying the reaction rate (i.e., the diffusion of the diamine) using the interlayers and the additives in the aqueous phase. The current FT-IR data would then be sufficient to illustrate the effects of the different interlayers and additives.
Otherwise, the manuscript still needs significant improvements in the discussion of the FT-IR results and possibly more data to explain how this characterization method is a reliable tool to study the IP reaction and how it can be used in the design of membranes like the author proposes.
Other minor comments:
3. The author followed my suggestion of using the word “pristine” to describe the polysulfone support without an interlayer. However, there are still some instances in the manuscript where the term pristine is used when discussing the membranes having an interlayer (e.g., line 266 or the legend of Figure 5). Please check the rest of the document.
4. There is no Figure S10 in the Supporting Information. Check the numbering of the figures and the corresponding references in the main text.
5. The manuscript still needs to be revised for language and grammar issues.
Author Response
Response to Reviewer 1 Comments
Point 1: In my previous report, I mentioned two significant issues that the author needed to correct before this manuscript could be accepted for publication:
According to the author, the main objective of this manuscript is to “monitor the IP reaction and the polyamide film growth on the porous substrate in real-time”. I recognized that Figure 2 showed that FT-IR was qualitatively sensitive to both the appearance of a polyamide layer on the support and to the effects of slowing down the reaction with the use of interlayers and additives. However, I was not entirely convinced that this data could be used to measure the thickness of the layers, and even less predict or describe the relative performance of the resulting membranes.The revised manuscript did not address any of these concerns, and in my opinion the same issues remain:
The results did not provide clear evidence of whether FT-IR was capable of reliably measuring the thickness of the layers under different reaction conditions. I was also expecting to see a better comparison (e.g., in a table) of the thickness values obtained with FT-IR vs. those obtained from the TEM images, as well as a detailed discussion of the shortcomings of each technique. Also, in order to validate the thickness calibration curves, images (or some other evidence) would be necessary at different reaction times to prove that the FT-IR values are truly tracking the growth of the layers. For example, if the standard reaction shown in the first panel of Figure 2 is allowed to proceed to 300 s, would the resulting polyamide layer be ~250 nm?
1b. There is no evidence or proper discussion of how this FT-IR method can actually help in the preparation of membranes with improved properties. The FT-IR data neither was used to optimize any of the membranes, nor provided essential information beyond the two qualitative aspects that I mentioned before. The change in the amine diffusivities was measured with UV-Vis and the relative thickness differences were shown with TEM images. In other words, the same membrane results would have been obtained without using FT-IR. Of course, it would be a significant result if this FT-IR method could predict the characteristics of the resulting polyamide layer, even if it is just the film thickness, but for this it would be necessary to establish and verify proper calibration curves.
1c. The revised version of Figure 2 introduced (in the first graph panel) a division of three reaction stages based on the mathematical modeling of the IP reaction by Freger and Srebnik (Ref. 36). However, I do not see any evidence that the FT-IR curve corresponds to these theoretical stages. The current data and experiments are not enough to investigate the existence of these three stages. Even the experimental work by Freger (Ref. 37) or any other study, as far as I know, have not shown this conclusively. Furthermore, such a formation mechanism would still be true at least for some of the other cases with different reaction rates. It would be necessary for the author to explain what exactly is FT-IR measuring in the other cases with respect to the formation of the polyamide layer.
Thanks for the review’s suggestion and advice. The article originally aims to monitor the in-situ polyamide formation by FT-IR spectroscopy, as a result, we can get the polyamide formation and simply transformed the FT-IR absorbance into the thickness of polyamide layer. However, I have lost many important messages in the previously manuscript. As I mentioned in this revised manuscript: “In addition, measuring the polyamide layer thicknesses through FT-IR in reference literature Ref S3 in the Supporting Information was only demonstrated for dense uniform films and although this is not the case for IP-based membranes, it could still yield valid information on the IP process. In the calculation in the Supporting Information, it is needed to assume the polymer density to be 1.5 even tough while this is approximate because the density is not constant throughout a PA film nor throughout the interfacial polymerization reaction time......Here, I take the carbonyl vibration at 1640 cm-1. However, this is the carbonyl vibration of the amide and that as such there will be an error depending on the variation in chemical structure which is based on the following XPS analysis, because polyamide structure is heterogeneous in the depth......As a result, this method of FT-IR spectroscopy does not take variations in surface morphology of the polyamide into account and that that is another potential source of error on the thickness determination using FT-IR spectroscopy. Because there are questions regarding the vibration, the chemistry, the density and in fact, I do not have a validation on the FT-IR based thickness, it would be better to use a unit-less thickness in Figure 3”. As a result, I am afraid of the main role of FT-IR spectroscopy is to monitor the interfacial polymerization process on the porous substrate. While it is difficult for directly polyamide layer thickness determination, and the thickness is approximate and hard to be compared with the layer thickness observed from TEM images. Thanks for the reviewer’s valuable advice. Actually I cannot demonstrate the 3-phase process well, because IP process is very complicated and I do not have enough evidence. As a result, I remove discussion about the 3-phase process and re-change the Figure 2 as the review suggested.
My second point was related to how the comparison of the membrane performance results were presented in a very misleading way in the original version of Figure 5. The updated version of Figure 5 made the necessary corrections and it addresses my concerns. The only minor comment is that the leftover red arrow in Figure 5b should be removed.
Thanks for the review’s advice. Figure 5b has been modified in the revised manuscript. The left-over red arrow in Figure 5b has been removed.
I would be inclined to support this manuscript if the author were to shift the main focus from a paper about FT-IR as a method to monitor the IP reaction to a paper about the fabrication of membranes with improved performance by modifying the reaction rate (i.e., the diffusion of the diamine) using the interlayers and the additives in the aqueous phase. The current FT-IR data would then be sufficient to illustrate the effects of the different interlayers and additives. Otherwise, the manuscript still needs significant improvements in the discussion of the FT-IR results and possibly more data to explain how this characterization method is a reliable tool to study the IP reaction and how it can be used in the design of membranes like the author proposes.
I totally agree with the review’s intelligent advice. Because I do not have enough experimental evidence to support this article on the FT-IR spectroscopy as a method to monitor the IP reaction, the main idea should be changed to the focus on fabrication of membranes with improved performance by modifying the reaction rate (using the interlayers and the additives in the aqueous phase), which I also think will be more persuasive, acceptable and easily to be understand for the readers.
The author followed my suggestion of using the word “pristine” to describe the polysulfone support without an interlayer. However, there are still some instances in the manuscript where the term pristine is used when discussing the membranes having an interlayer (e.g., line 266 or the legend of Figure 5). Please check the rest of the document.
Thanks for the review’s advice. The term “pristine” has been all checked in the rest of the revised document of the manuscript.
4. There is no Figure S10 in the Supporting Information. Check the numbering of the figures and the corresponding references in the main text.
I am sorry for the missing of Figure S10 in the Supporting Information, It has been modified in the revised manuscript as Figure S12 in the Supporting Information.
5. The manuscript still needs to be revised for language and grammar issues.
The revised manuscript has been revised for language and grammar issues, thanks the review’s advice and great patience.

Reviewer 2 Report
I would like to thank the author for the implementation of many of the requested changes.
I do however still have 2 major issues that, for me, are still not improved drastically enough to suggest accepting this paper.
(1) With respect to the IR measurements ('point2'). This was my main point why i suggested rejection of the manuscript. In my opinion what I said is still a valid concern and you do not demonstrate a critical attitude towards these experiments in the revised manuscript.
I do want to give constructive feedback so let me suggest what I think is necessary. The main message is that you should critically analyse both the experimental limitations of an experiment/sample and the generated data. This critical analysis should be discussed in the manuscript.
You need to mention in the manuscript that measuring thicknesses through FTIR in literature was only demonstrated for dense uniform films and although this is not the case for IP-based membranes, it could still yield valid information on the IP-process. You need to mention that you assume the density to be 1.5 even tough you know that this is not correct as you know the density is not constant throughout a PA-film nor throughout time. You mention that you take the carbonyl vibration at 1640 cm-1. You additionally need to mention that this is the carbonyl vibration of the amide and that as such there will be an error depending on the variation in chemical structure that you know is there based on your XPS. You need to mention that this method does not take variations in surface morphology into account and that that is another potential source of error on the thickness determination using FTIR. Because there are so many questions regarding the vibration, the chemistry, the density and in fact you do not have a validation on the FTIR-based thickness, I wonder if it would not be better to use a unitless thickness in figure 3. I don't think the use of 'nm' is warranted given that there is the experimental uncertainty is so big.
These 5 remarks all have to be implemented in the manuscript's discussion before I think the paper can even be considered to be published.
My second remaining remark is regarding the 3-phase process in figure 2. It is nice that you added the 3 phases to the graph which makes it clearer what you mean but I still do not see it... you cannot just randomly put 3 phases on there. I can see that you want to see a kind of exponential phase in the beginning and a more linear phase after that, but i don't see why you would see a 3rd phase? I just don't see it. If you cannot demonstrate it, please remove all talk on the 3-phase process.
Author Response
Response to Reviewer 2 Comments
Point 1: I would like to thank the author for the implementation of many of the requested changes.
I do however still have 2 major issues that, for me, are still not improved drastically enough to suggest accepting this paper.
(1) With respect to the IR measurements ('point 2'). This was my main point why I suggested rejection of the manuscript. In my opinion what I said is still a valid concern and you do not demonstrate a critical attitude towards these experiments in the revised manuscript. I do want to give constructive feedback so let me suggest what I think is necessary. The main message is that you should critically analyse both the experimental limitations of an experiment/sample and the generated data. This critical analysis should be discussed in the manuscript.
You need to mention in the manuscript that measuring thicknesses through FT-IR in literature was only demonstrated for dense uniform films and although this is not the case for IP-based membranes, it could still yield valid information on the IP process. You need to mention that you assume the density to be 1.5 even tough you know that this is not correct as you know the density is not constant throughout a PA-film nor throughout time. You mention that you take the carbonyl vibration at 1640 cm-1. You additionally need to mention that this is the carbonyl vibration of the amide and that as such there will be an error depending on the variation in chemical structure that you know is there based on your XPS. You need to mention that this method does not take variations in surface morphology into account and that that is another potential source of error on the thickness determination using FT-IR. Because there are so many questions regarding the vibration, the chemistry, the density and in fact you do not have a validation on the FT-IR based thickness, I wonder if it would not be better to use a unitless thickness in figure 3. I don't think the use of 'nm' is warranted given that there is the experimental uncertainty is so big.
These 5 remarks all have to be implemented in the manuscript's discussion before I think the paper can even be considered to be published.
Thanks for the reviewer’s valuable advice and great patience. These 5 remarks you mentioned above all have been modified in the revised manuscript.
Point 2: My second remaining remark is regarding the 3-phase process in figure 2. It is nice that you added the 3 phases to the graph which makes it clearer what you mean but I still do not see it... you cannot just randomly put 3 phases on there. I can see that you want to see a kind of exponential phase in the beginning and a more linear phase after that, but i don't see why you would see a 3rd phase? I just don't see it. If you cannot demonstrate it, please remove all talk on the 3-phase process.
Thanks for the reviewer’s valuable advice. Actually I cannot demonstrate the 3-phase process well, because IP process is very complicated and I do not have enough evidence. As a result, I remove discussion about the 3-phase process and re-change the Figure 2 as the review suggested.

Round 3
Reviewer 1 Report
In this second revised version of the manuscript, the author provides a clearer presentation and discussion of the FT-IR results based on the data collected and how the experiments were designed. Thus, the initial concerns regarding unsupported claims have been addressed.
-Line 31: "As it is...
-Line 236: "that that"
Reviewer 2 Report
Thank you for taking my suggestions into account. I think these make the scientific results of the manuscript much more sound.